# Factors associated with the time to return negative RT-PCR from COVID-19 in paediatric patients: a retrospective cohort study

Jianbo Shao,[1] Hong Xu,[2] Zhixi Liu ![ORCID],[3] Xiaohua Ying,[4] Hua Xu,[5] Xianfeng Wang,[6] Jinmiao Lu,[7] Yidie Huang,[7] Guangfei Wang,[7] Yanling He,[8] Jie Chen,[9] Shuli Ma,[10] Shangrong Zou,[11] Yuxia Cui,[12] Ruijie Chen,[13] Jin Lu,[14] Xuyuan Li,[15] Zhiping Li,[7] Guoying Huang,[16] Weibing Wang ![ORCID] [3]

JS, HX, ZL and XY contributed equally.
ZL, GH and WW contributed equally.

JS, HX, ZL and XY are joint first authors.

For numbered affiliations see end of article.

**Correspondence to**
Professor Weibing Wang;
wwb@fudan.edu.cn,
Dr Guoying Huang;
gyhuang@shmu.edu.cn and
Dr Zhiping Li;
zpli@fudan.edu.cn

## ABSTRACT

**Objective** This study aimed to describe the epidemiological and clinical features and potential factors related to the time to return negative reverse transcriptase (RT)-PCR in discharged paediatric patients with COVID-19.

**Design** Retrospective cohort study.

**Setting** Unscheduled admissions to 12 tertiary hospitals in China.

**Participants** Two hundred and thirty-three clinical charts of paediatric patients with confirmed diagnosis of COVID-19 admitted from 1 January 2020 to 17 April 2020.

**Primary and secondary outcome measures** Primary outcome measures: factors associated with the time to return negative RT-PCR from COVID-19 in paediatric patients. Secondary outcome measures: epidemiological and clinical features and laboratory results in paediatric patients.

**Results** The median age of patients in our cohort was 7.50 (IQR: 2.92–12.17) years, and 133 (57.1%) patients were male. 42 (18.0%) patients were evaluated as asymptomatic, while 162 (69.5%) and 25 (10.7%) patients were classified as mild or moderate, respectively. In Cox regression analysis, longer time to negative RT-PCR was associated with the presence of confirmed infection in family members (HR (95% CI): 0.56 (0.41 to 0.79)). Paediatric patients with emesis symptom had a longer time to return negative (HR (95% CI): 0.33 (0.14 to 0.78)). During hospitalisation, the use of traditional Chinese medicine (TCM) and antiviral drugs at the same time is less conducive to return negative than antiviral drugs alone (HR (95% CI): 0.85 (0.64 to 1.13)).

**Conclusions** The mode of transmission might be a critical factor determining the disease severity of COVID-19. Patients with emesis symptom, complications or confirmed infection in family members may have longer healing time than others. However, there were no significant favourable effects from TCM when the patients have received antiviral treatment.

## Strengths and limitations of this study

► This study enrolled all paediatric patients with COVID-19 (n=233) discharged from 12 tertiary hospitals across 11 cities and found paediatric patients with emesis symptom had a longer time to return negative.

► We determined the in-family transmission is the main infection route for paediatric COVID-19 cases.

► Transmission mode may be a key factor in determining the clinical course of COVID-19.

worldwide has reached 203 944 144 with 121 960 of those cases in China.[1] On 11 March 2021, the WHO further announced the COVID-19 epidemic would be classified as a global pandemic.[2] According to the data released by the WHO,[3] pneumonia is the leading cause of death among children worldwide and the second among children under 5 years in China.[4] Paediatric cases were reported during the epidemics of SARS-CoV.[5 6] In paediatric patients with SARS, the clinical symptoms of teenagers were similar to those of adults.[7] Furthermore, they maintained stable conditions throughout the disease course and had good prognoses.[8] Another study demonstrated that paediatric cases accounted for only 2% of all Middle East respiratory syndrome cases.[9] One previous study that enrolled 36 COVID-19 paediatric cases concluded that the close contacts with family members might be difficult to avoid in daily life.[10]

The low proportion of paediatric patients among COVID-19 cases has perplexed clinicians, epidemiologists and scientists. Current understanding regarding the duration of positive reverse transcriptase (RT)-PCR to negative for COVID-19 is limited, and this

## INTRODUCTION

COVID-19 is an emerging infectious disease with high transmissibility. As of 11 August 2021, the cumulative number of cases

knowledge gap is particularly notable among children. Therefore, we aimed to explore and analyse the clinical and epidemiological characteristics of 233 discharged paediatric patients to reveal corresponding information for the prevention and treatment of paediatric patients with COVID-19.

## MATERIALS AND METHODS
### Study population
This multicentre retrospective study was conducted in 12 tertiary hospitals in 11 Chinese cities. We collected information about the discharged paediatric patients with COVID-19 treated in these hospitals from 1 January 2020 to 17 April 2020. The paediatric patients with confirmed COVID-19 were diagnosed based on guidelines issued by the National Health Commission of the People's Republic of China.[11]

### Procedure
Relevant medical information was collected from electronic medical records and reviewed by trained doctors to ensure the quality of the medical data. Retrieved information included the demographic information, COVID-19 contact history (eg, history of travel to or residence in the COVID-19 epidemic areas or history of contact with SARS-CoV-2-infected individuals), complications (eg, myocardial injury[12]: the elevation of cardiac troponin, with at least one value above the 99th percentile upper reference limit; liver insufficiency[13]: the inability of the liver to perform its normal synthetic and metabolic functions with abnormal values of biochemical indicators aspartate aminotransferase (AST), alanine aminotransferase (ALT), etc, the ranges of normal values (shown in table 1), laboratory tests results, chest CT and B-scan ultrasonography results. Laboratory tests included blood routine examination and examinations for lung, liver, myocardial and kidney function, such as measurements of the AST, ALT, lactate dehydrogenase (LDH), α-hydroxybutyrate dehydrogenase (α-HBDH) and erythrocyte sedimentation rate (ESR).

The preliminary diagnosis of suspected COVID-19 was based on clinical symptoms and potential exposure via travel to COVID-19 epidemic areas or close contact with a confirmed case. The diagnosis was subsequently confirmed based on results from real-time RT-PCR tests for SARS-CoV-2 on nasal or pharyngeal swab specimens from all suspected patients diagnosed by the previous step. RT-PCR is recommended by the Centers for Disease Control and Prevention[11] and WHO interim guidance as a solid method for confirming COVID-19 diagnosis.[14] RT-PCR testing was performed to detect infection with other viruses (ie, adenovirus, influenza virus, parainfluenza virus and respiratory syncytial virus) and sputum culture was performed to test for potential infections with bacteria, including *Legionella pneumophila*, *Mycoplasma pneumoniae* and *Chlamydia pneumoniae*. We evaluated the clinical type of each admitted paediatric patient with COVID-19 according to the recommendations issued by the paediatrics branch of the Chinese Medical Association.[15] Patients were described as having one of four clinical types of COVID-19, that is, asymptomatic, mild, moderate and severe disease, based on their RT-PCR results, clinical symptoms and radiological results (online supplemental table 1).[16 17] Once admitted, these paediatric patients were given available antiviral treatments (eg, ribavirin aerosol, oseltamivir granule, etc) and oxygen if necessary (blood oxygen saturation <0.92). The discharge criteria were that the patient must have obtained two consecutive negative RT-PCR test results.

### Statistical analysis
Categorical variables are presented as number and frequency rate using the Fisher's exact test or $X^2$ test. Continuous variables with non-normal distribution are presented as the median and IQR and were compared using the Kruskal-Wallis rank-sum test. In addition, we analysed the first and last laboratory test results to examine the effect of treatment during hospitalisation. These two test results were compared by a paired Student's t-test or Wilcoxon signed-rank test.

Cox proportional hazards regression models were used to estimate HRs and corresponding 95% CIs to explore associations between risk factors and the number of days to return negative RT-PCR for the 233 paediatric COVID-19 cases. In Cox regression models, the interpretation of results is that the endpoint event we observed was negative RT-PCR (benign event); if the HR value of a variable is less than 1, the variable is not conducive to the occurrence of the outcome of negative RT-PCR, and there will be a longer day to return negative, and vice versa. Time to return negative was defined as the time from positive RT-PCR results to negative RT-PCR results for SARS-CoV-2 during the hospitalisation. During the analysis, sample size varied because of missing data; the sample sizes for each analysis are listed in table 2. The variables assessed in the statistical analysis were contact history, clinical symptoms and clinical indicators. Univariate and multivariate analyses adjusted for sex, age and weight were carried out by Cox proportional HR models to examine significant associations. These analyses conducted by R packages 'dplyr', a data manipulation package, will provide any information about factors that contribute to longer duration of positive RT-PCR to negative; 'survival' and 'survminer' helped to determine the risk factors that contributed to a longer time of positive RT-PCR.

All analyses were conducted on the subjects who lacked missing data, and only differences with two-sided α and a p value less than 0.05 were considered statistically significant. All analyses were conducted with the use of R software V.3.6.2.

### Patient and public involvement
Patients and/or the public were not involved in the design, or conduct, or reporting or dissemination plans of this research.

**Table 1** Laboratory indices at hospital admission of paediatric patients with COVID-19 pneumonia

| Tests in study population | Reference values | No of patients tested | Median (IQR) | No of patients with value deviation from reference (%) | |
|---|---|---|---|---|---|
| | | | | Below reference | Above reference |
| **Haematological** | | | | | |
| White cell count, ×10⁹/L | Varied with age* | 229 | 6.64 (5.26–8.18) | 50 (22.0) | 17 (7.4) |
| Lymphocytes, % | Varied with age* | 229 | 42.40 (33.00–55.10) | 52 (22.7) | 98 (42.8) |
| Platelets, ×10⁹/L | 100–400 | 224 | 266.00 (225.00–325.00) | 1 (0.4) | 20 (8.9) |
| Haemoglobin, g/L | Varied with age* | 224 | 129.00 (119.80–138.20) | 16 (7.2) | 7 (3.2) |
| **Biochemical** | | | | | |
| AST, IU/L | 13–35 | 225 | 29.00 (20.00–41.00) | 10 (4.4) | 69 (30.7) |
| ALT, IU/L | 7–40 | 227 | 15.50 (11.00–24.75) | 16 (7.0) | 20 (8.8) |
| Albumin, g/L | 40–55 | 223 | 45.60 (43.10–48.05) | 16 (7.2) | 3 (1.3) |
| Prealbumin, mg/L | 200–400 | 142 | 194.30 (153.85–229.68) | 76 (53.5) | – |
| Urea, mmol/L | 2.8–7.6 | 218 | 3.97 (3.23–5.06) | 28 (12.8) | 11 (5.0) |
| Creatinine, μmol/L | Varied with age* and sex | 225 | 36.60 (26.80–48.90) | 8 (3.6) | 12 (5.4) |
| Cholinesterase, U/L | 5300–11300 | 40 | 8914.00 (7690.00–10022.00) | – | 4 (10.0) |
| LDH, U/L | Varied with age* | 196 | 238.00 (194.80–311.80) | 2 (1.0) | 36 (18.3) |
| α-HBDH, U/L | 72–182 | 22 | 185.50 (162.50–231.50) | – | 13 (59.1) |
| Procalcitonin, ng/mL | <0.1 | 186 | 0.05 (0.03–0.08) | – | 41 (20.2) |
| **Infection-related indices** | | | | | |
| IL-6, pg/mL | <7 | 145 | 3.63 (2.36–6.85) | – | 35 (24.1) |
| CRP, mg/L | <8 | 211 | – | – | 40 (19.0) |
| ESR, mm/hour | Varied with age and sex† | 112 | 10.00 (5.75–19.00) | – | 19 (17.0) |
| **Coagulation function** | | | | | |
| PT, s | 11–14.5 | 191 | 11.40 (10.80–12.50) | 59 (30.9) | 6 (3.1) |
| APTT, s | 26–40 | 191 | 32.8 (29.70–35.35) | 12 (6.3) | 25 (13.1) |
| D-dimer, μg/mL | 0–0.5 | 150 | 0.26 (0.15–0.42) | – | 28 (18.7) |
| Serum troponin, μg/L | <0.01 | 55 | – | – | 18 (30.0) |

*White cell count (×10⁹/L): <28 days old, 10.0–24.0; 29 days–3 years old, 8.0–12.0; >3 years old, 4.0–10.0. Lymphocytes (%): <28 days old, 30–40; 29 days–3 years old, 50–70; >3 years old, 30–40. Haemoglobin (g/L): <28 days old, 10.0–24.0; 29 days–3 years, 8.0–12.0; >3 years old, 4.0–10.0. LDH: 0–29 days old, 290–2000; 30 days–23 months old, 180–430; >23 months old, 110–290.
†Creatinine (μmol/L): ≤2 months old, 22–90; 2 months–3 years old, 11–34; 3–15 years old, 21–65; >15 years old (male), 64–104, (female), 49–90. ESR (mm/hour) (male): 0–21, (female): 0–26.
ALT, alanine aminotransferase; APTT, activated partial thromboplastin time; AST, aspartate aminotransferase; CRP, C reactive protein; ESR, erythrocyte sedimentation rate; α-HBDH, α-hydroxybutyrate dehydrogenase; IL-6, interleukin 6; LDH, lactate dehydrogenase; PT, prothrombin time.

## RESULTS
### Demographic and clinical characteristics
From 1 January 2020 and 17 April 2020, 233 paediatric COVID-19 cases were discharged from the 12 study hospitals (online supplemental figure 1). These cases with a median age of 7.50 (IQR: 2.92–12.17) years ranged from 0.05 to 18.75 years old of which 133 (57.1%) cases were male patients. Most patients (n=199 (85.4%)) had a COVID-19 contact history, including contact with people from Wuhan/Hubei or travel to Wuhan/Hubei. Fifty-nine (26.5%) of the paediatric patients were the first

SARS-CoV-2-infected individual of their family. Additionally, 40 (17.2%) patients had a simultaneous infection with an additional virus, and 79 (33.9%) patients had a simultaneous bacterial infection. None of the paediatric patients in our study were found to be at risk of death due to COVID-19 during hospitalisation (table 2).

The most common clinical symptoms of the 233 paediatric patients with COVID-19 were fever (36.9%), cough (35.6%) and nausea (16.3%). The symptoms of runny nose, diarrhoea and pharyngitis were also reported. All patients underwent a chest X-ray or CT test, and the

**Table 2** Clinical features of 233 paediatric patients infected with SARS-CoV-2

| Clinical factors | | All patients N (%) | Asymptomatic N (%) | Mild* N (%) | Moderate† severe N (%) | P value‡ |
|---|---|---|---|---|---|---|
| | | 233 (100.0) | 42 (18.0) | 162 (69.5) | 29 (12.4) | |
| Sex | Male | 133 (57.1) | 29 (69.0) | 88 (54.3) | 16 (55.2) | 0.223 |
| | Female | 100 (42.9) | 13 (31.0) | 74 (45.7) | 13 (44.8) | |
| Age, years | ≤5 | 84 (36.1) | 6 (14.3) | 61 (37.7) | 17 (58.6) | <0.001 |
| | >5 | 149 (63.9) | 36 (85.7) | 101 (62.3) | 12 (41.4) | |
| Age, median (IQR), years | | 7.50 (2.92–12.17) | 9.37 (7.00–12.15) | 7.92 (2.85–12.29) | 4.00 (0.85–8.92) | 0.010§ |
| Contact history of epidemic area | Yes | 199 (85.4) | 41 (97.6) | 132 (81.5) | 26 (89.7) | 0.015¶ |
| Number of infected family member | 0 | 59 (26.5) | 4 (9.8) | 46 (29.5) | 9 (34.6) | 0.185 |
| | 1 | 67 (30.0) | 16 (39.0) | 43 (27.6) | 8 (30.8) | |
| | 2 | 54 (24.2) | 10 (24.4) | 39 (25.0) | 5 (19.2) | |
| | ≥3 | 43 (19.3) | 11 (26.8) | 28 (17.9) | 4 (15.4) | |
| Combined with virus infection** | Yes | 40 (17.2) | 2 (4.8) | 31 (19.1) | 7 (24.1) | 0.039 |
| Combined with bacterial infection†† | Yes | 79 (33.9) | 13 (31.0) | 60 (37.0) | 6 (20.7) | 0.142 |
| Fever | Yes | 86 (36.9) | 0 (0.0) | 69 (42.6) | 17 (58.6) | <0.001 |
| Cough | Yes | 83 (35.6) | 0 (0.0) | 68 (42.0) | 15 (51.7) | <0.001 |
| Runny nose | Yes | 21 (9.0) | 0 (0.0) | 17 (10.5) | 4 (13.8) | 0.027¶ |
| Diarrhoea | Yes | 12 (5.2) | 0 (0.0) | 9 (5.6) | 3 (10.3) | 0.127¶ |
| Nausea | Yes | 38 (16.3) | 0 (0.0) | 32 (19.8) | 6 (20.7) | 0.001¶ |
| Coughing of phlegm | Yes | 23 (9.9) | 0 (0.0) | 16 (9.9) | 7 (24.1) | 0.002¶ |
| Pharyngitis | Yes | 13 (5.6) | 0 (0.0) | 10 (6.2) | 3 (10.3) | 0.111¶ |
| X-ray or CT test findings | Abnormal | 139 (59.7) | 17 (40.5) | 94 (58.0) | 28 (96.6) | <0.001 |
| Complications | Yes | 31 (13.3) | 3 (7.1) | 22 (13.6) | 6 (20.7) | 0.268 |
| Days in hospital, median (IQR) | | 12.00 (9.00–17.00) | 10.50 (8.25–14.00) | 12.00 (9.00–17.00) | 13.00 (9.75–17.00) | 0.416§ |

*Mild disease: upper respiratory symptoms for short duration or asymptomatic infection; positive reverse transcriptase-PCR test for SARS-CoV-2; no abnormal radiographic and septic presentation.
†Moderate disease: mild pneumonia; symptoms such as fever, cough, fatigue, headache and myalgia; no complications and manifestations related to severe conditions.
‡P values indicate the difference among four different disease types in paediatric patients with COVID-19.
§P value was obtained from a Kruskal-Wallis test.
¶P values were obtained from Fisher's exact tests.
**Adenovirus, influenza virus, parainfluenza virus, respiratory syncytial virus, etc.
††*Legionella pneumophila, Mycoplasma pneumoniae, Chlamydia pneumoniae*, etc.

results suggest that 139 (59.7%) patients had lung abnormalities, which include 17 (40.5%) of asymptomatic patients, 94 (58.0%) of mild patients, and 28 (96.6%) of moderate or severe patients. The lung abnormalities included ground-glass opacities and patchy shadows or multiple plaques in some of the lung lobes on a CT scan. The median length of the patient hospital stay was 12 days (IQR: 9–17 days). An examination of patients' complications revealed that 31 of the 233 paediatric patients with COVID-19 had complications (eg, myocardial injury, liver insufficiency, etc), including 3 (7.1%) asymptomatic patients, 22 (13.6%) mild patients, and 6 (20.7%) moderate or severe patients.

The clinical disease was classified as asymptomatic in 42 (18.0%) patients, mild in 162 (69.5%) patients, moderate in 25 (10.7%) patients and severe in 4 (1.7%) patients. As the severity of disease increased, the median age (p=0.010) decreased and the proportion of children under 5 years old (p<0.001) increased. Notably, there was no significant sex difference in severity of disease (p=0.223). When comparing between children with or without infected family members, the severity of illness exhibited no significant correlation. Among the patients with moderate or severe disease, 17 (58.6%) had fever, 15 (51.7%) had cough and 7 (24.1%) had coughing of phlegm. Only 13.8% and 10.3% had the symptoms of runny nose and diarrhoea, respectively. Of the mild cases, 69 (42.6%), 68 (42.0%), and 32 (19.8%) patients had fever, cough, and nausea, respectively. In contrast, none of the asymptomatic patients had any of the clinical symptoms that were common in the patients with mild or moderate disease.

## Laboratory indices at hospital admission for the 233 paediatric patients with COVID-19

Here are the results of laboratory indices obtained in the first laboratory test after hospital admission (table 1). Of the 229 cases, 50 had leucopenia. More than half of the patients (n=150 (65.50%)) had an abnormal lymphocyte percentage; 52 (22.70%) had lymphopenia. In addition, some patients were diagnosed with abnormal liver function, as indicated by 69 (30.70%) patients with elevated AST levels, while 20 (8.80%) patients had elevated ALT levels and 41 (20.20%) had increased procalcitonin levels. We also detected renal damage in some patients, as indicated by elevated levels of creatinine (n=12 (5.40%)) and urea (n=11 (5.00%)). Some patients exhibited elevated levels of LDH (n=36 (18.30%)) and α-HBDH (n=13 (59.10%)). Levels of infection-related indices, such as interleukin 6 (IL-6) (35 (24.10%)), C reactive protein (CRP) (40 (19.00%)) and ESR (19 (17.00%)), were also elevated in some patients.

Figure 1 shows the difference between the results of the first and last laboratory tests for the 233 paediatric COVID-19 cases. The value of liver damage indices (AST (difference: 6.05 IU/L; 95% CI: 1.96 to 10.14 IU/L; p=0.004)) and myocardial indices (LDH (difference: 40.33 IU/L; 95% CI: 5.11 to 75.55 IU/L; p=0.026)) were significantly decreased over the disease course. Additionally, the lymphocyte percentage (difference:

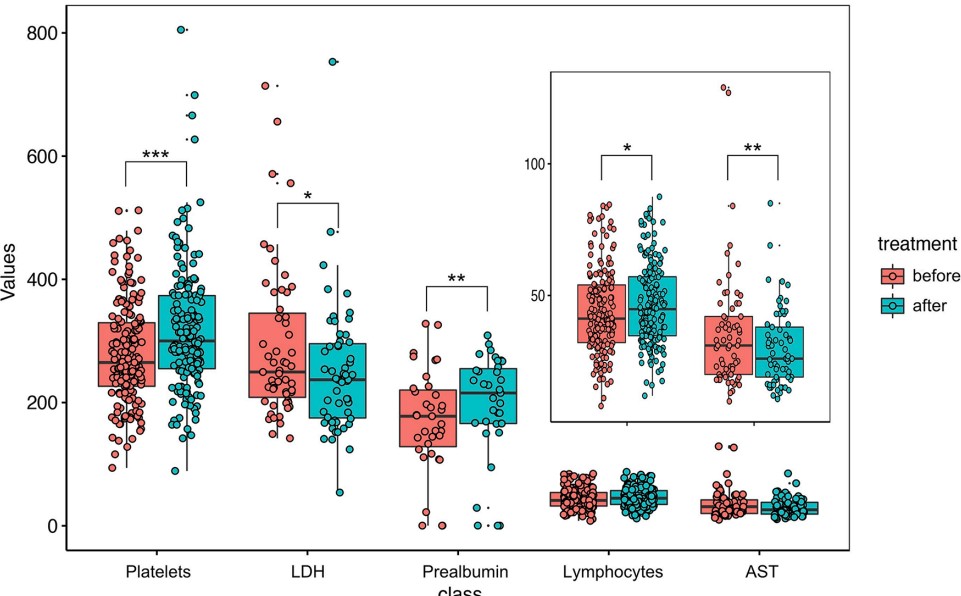

**Figure 1** Comparing the clinical indicators of paediatric patients with COVID-19 pre-therapy and post-treatment during hospitalisation. The effect of treatment in 233 paediatric patients with COVID-19. The red and green boxes represent the values of five clinical indicators before and after treatment, respectively. The scattered points on each box represent the values of each clinical indicator for each patient. The middle line of the box represents the medium value of the group. The height of box represents the IQR of each group. The P value is calculated based on the paired Student's t-test. *Represents a p value of less than 0.05; **represents a p value of less than 0.005; ***represents a p value of less than 0.001. AST, aspartate aminotransferase; LDH, lactate dehydrogenase.

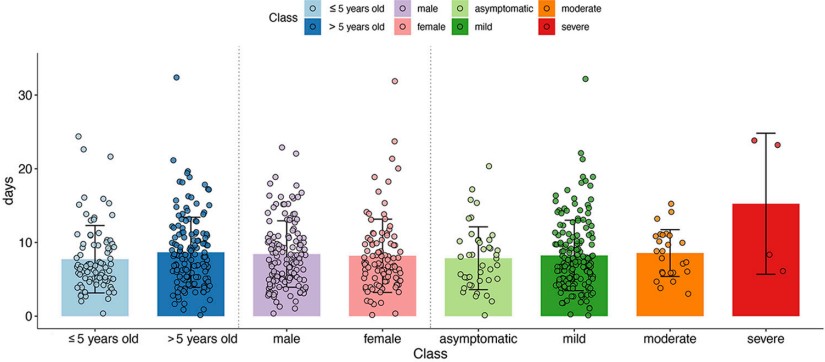

**Figure 2** Distribution of the time to return negative of the 233 paediatric patients with COVID-19. The distribution of the mean time to return negative reverse transcriptase (RT)-PCR for the 233 paediatric patients with COVID-19 during the period of admission, split in groups based on different classification criteria. The gradient blue bars represent the mean number of days to return negative at age ≤5 years and age >5 years. The purple and pink bars represent mean number of days to return negative for male and female patients, respectively. The four bars on the right represent the mean number of days to return negative with the four clinical types of COVID-19. The scattered points on each bar represent the number of days to return negative for each patient in the group. The error lines represent the mean value plus or minus an SD. Turning negative was defined as the time at which a patient with confirmed COVID-19 first received negative results for an RT-PCR test for SARS-CoV-2.

−2.82%; 95% CI: −4.79% to −0.85%; p<0.006), platelet count (difference: −38.74×$10^9$/L; 95% CI: −53.46 to −24.02×$10^9$/L; p<0.0001) and prealbumin levels (difference: −25.15 mg/L; 95% CI: −39.41 to −10.89 mg/L; p=0.001) were significantly increased by the final laboratory test. In order to remove age effects, the study examined the trends of different clinical indicators before and after treatment by age stratification (online supplemental figure 2).

### Factors impacting the duration to return negative RT-PCR

Figure 2 shows the distribution in the number of days to return negative (ie, the number of days from positive RT-PCR to negative RT-PCR results for SARS-CoV-2) in the 233 paediatric COVID-19 cases based on different classification criteria, including age, sex and clinical type. This length of duration time from positive RT-PCR to negative for individual patients ranged from 0 to 32 days. The mean lengths of time to return negative RT-PCR for asymptomatic, mild, moderate, and severe cases were 7.86±4.26, 8.24±4.77, 8.56±3.18, and 15.25±9.57 days, respectively. Regarding patient age, the mean numbers of days to return negative were 7.73±7.73 and 8.66±4.76 for the groups aged less than 5 years old and more than 5 years old, respectively. Regarding patient sex, the mean lengths of time to return negative were 8.43±4.51 and 8.19±4.97 days in male and female patients, respectively.

We constructed Cox proportional hazards regression models to explore the factors that influenced the time to return negative. The results are shown in table 3. After controlling for the factors of sex, age and weight, comparing the HRs for the time from positive RT-PCR to negative RT-PCR between patients who had infected family members and those who did not have infected family members yielded an HR and corresponding 95% CI of 0.56 (0.41 to 0.79). Furthermore, comparisons of patients who had zero infected family members with those who

had one, two, or at least three infected family members yielded HRs (95% CI) of 0.47 (0.32 to 0.69), 0.51 (0.34 to 0.76), and 0.56 (0.36 to 0.86), respectively; and a lower HR means a longer time to return negative. Regarding the clinical disease types, we found the group with severe disease had a significantly longer time to return negative RT-PCR (HR (95% CI): 0.14 (0.03 to 0.59)), after controlling for demographic factors (p=0.008), while the time from positive RT-PCR to negative of mild and moderate patients was not significantly different from that of asymptomatic patients. The symptom of the occurrence of emesis was significantly associated with the time to return negative (HR (95% CI): 0.33 (0.14 to 0.78)), as was having complications (HR (95% CI): 0.57 (0.37 to 0.89)). There was no significant association between use of traditional Chinese medicine (eg, Forsythia, Honeysuckle, Isatis root) and antiviral drug and time to return negative (HR (95% CI): 0.85 (0.64 to 1.13)).

### DISCUSSION

We found that nearly 70.4% of the paediatric COVID-19 cases were infected through close contact with family members. The paediatric patients with COVID-19 in our cohort were diagnosed with mainly mild disease, followed by asymptomatic and moderate disease. The proportion of severe patients was lower compared with that generally observed in adult patients, a pattern which is consistent with other studies.[18] [19] The median length of hospital stay in our paediatric patients with COVID-19 was 12 days, which is similar to that reported for a paediatric COVID-19 case study in Wuhan.[20] Paediatric patients with COVID-19 generally exhibited the clinical symptoms of fever, cough, nausea and coughing of phlegm, which was consistent with other studies.[21] It has been reported that adult patients with gastrointestinal symptoms are more

**Table 3** HRs of time to return negative reverse transcriptase-PCR from COVID-19 in Cox proportional hazards regression model

| Risk factors | Univariate Cox regression* | | Multivariate Cox regression*† | |
|---|---|---|---|---|
| | HR (95% CI) | P value | HR (95% CI) | P value |
| Clinical characteristics | | | | |
| Sex (vs male) | 1.10 (0.81 to 1.37) | 0.702 | – | – |
| Age | 0.82 (0.63 to 1.08) | 0.160 | – | – |
| Contact history | 1.40 (0.96 to 2.02) | 0.079 | 1.02 (0.67 to 1.57) | 0.918 |
| Clinical symptoms | | | | |
| Fever | 0.94 (0.72 to 1.23) | 0.655 | 0.96 (0.72 to 1.28) | 0.789 |
| Cold | 1.10 (0.27 to 4.45) | 0.890 | 1.15 (0.28 to 4.67) | 0.844 |
| Diarrhoea | 1.70 (0.97 to 3.13) | 0.063 | 1.56 (0.85 to 2.90) | 0.155 |
| Emesis | 0.51 (0.24 to 1.11) | 0.089 | 0.33 (0.14 to 0.78) | 0.012 |
| Nausea | 0.85 (0.60 to 1.20) | 0.348 | 0.76 (0.53 to 1.10) | 0.147 |
| Runny nose | 0.98 (0.63 to 1.54) | 0.934 | 0.94 (0.58 to 1.53) | 0.814 |
| Coughing of phlegm | 0.94 (0.61 to 1.45) | 0.778 | 0.90 (0.57 to 1.43) | 0.658 |
| Complications | 0.74 (0.50 to 1.08) | 0.12 | 0.57 (0.37 to 0.89) | 0.013 |
| Clinical type (vs asymptomatic) | | | | |
| Mild | 0.91 (0.65 to 1.28) | 0.596 | 0.92 (0.65 to 1.31) | 0.663 |
| Moderate | 0.92 (0.56 to 1.51) | 0.743 | 0.86 (0.50 to 1.48) | 0.591 |
| Severe | 0.29 (0.10 to 0.86) | 0.025 | 0.14 (0.03 to 0.59) | 0.008 |
| Household contact | | | | |
| Family member infection | 0.65 (0.48 to 0.89) | 0.006 | 0.56 (0.41 to 0.79) | 0.001 |
| Number of infected family member | | | | |
| 1 | 0.56 (0.39 to 0.80) | 0.002 | 0.47 (0.32 to 0.69) | <0.001 |
| 2 | 0.62 (0.43 to 0.90) | 0.012 | 0.51 (0.34 to 0.76) | 0.001 |
| ≥3 | 0.61 (0.41 to 0.92) | 0.019 | 0.56 (0.36 to 0.86) | 0.008 |
| Other respiratory pathogen infections | | | | |
| Combined with virus infection‡ | 0.92 (0.65 to 1.30) | 0.621 | 0.84 (0.59 to 1.21) | 0.348 |
| Combined with bacterial infection§ | 1.10 (0.82 to 1.44) | 0.574 | 1.04 (0.78 to 1.40) | 0.771 |
| Treatment (vs ATD) | | | | |
| TCM +ATD | 0.80 (0.61 to 1.04) | 0.093 | 0.85 (0.64 to 1.13) | 0.259 |

*Results of Cox proportional hazards regression models are represented as HRs and 95% CIs. Wald tests were performed to calculate the p values for trend.

†Adjusted for sex, age and weight.

‡Other detected viruses included adenovirus, influenza virus, parainfluenza virus and respiratory syncytial virus.

§Detected bacteria included *Legionella pneumophila*, *Mycoplasma pneumoniae* and *Chlamydia pneumoniae*.

ATD, antiviral drug; TCM, traditional Chinese medicine.

likely to develop severe conditions[22] which is consistent with the finding in paediatric patients. Notably, the paediatric COVID-19 cases who exhibited emesis symptom took a longer time to be clear from detectable SARS-CoV-2 (HR (95% CI): 0.33 (0.14 to 0.78)).

In contrast with a similar study,[10] we did find 17 asymptomatic paediatric patients with COVID-19 who had abnormal X-ray or CT test results. Although the asymptomatic paediatric patients with COVID-19 did not present the typical symptoms of pneumonia, the radiographic examinations were still able to provide some supportive evidence for diagnosis. Since most paediatric patients with COVID-19 have mild-type disease, routine CT examination is necessary to detect it because a plain X-ray examination may fail to show some lesion details, leading to a misdiagnosis or missed diagnosis.[23] We observed a tendency toward a higher detection of abnormalities in the lungs by X-ray or CT with increasingly more severe disease, evidenced by the detection of abnormalities in radiographic examinations for all the patients with moderate or severe-type disease in our study (p<0.001). Additionally, younger paediatric patients with COVID-19 were found to be more prone to developing severe conditions (p<0.001). Therefore, if they become

infected with SARS-CoV-2, they would be more prone to developing a more severe condition with a more fragile immune system.

A small proportion of paediatric patients with COVID-19 were found to have complications, although the incidence of complications in children with COVID-19 was lower than that observed in adults. Paediatric patients with COVID-19 with underlying medical conditions were detected to have some abnormal clinical indices (eg, elevated AST and procalcitonin levels and longer prothrombin time), and they took longer to return negative RT-PCR than did those without any complications (HR (95% CI): 0.57 (0.37 to 0.89)). We suspected that paediatric patients with COVID-19 whose physical functionality may be impaired before infection with SARS-CoV-2, for example, by an underlying infection or other complications, will need more time to return negative from COVID-19.

Elevated levels of inflammatory factors such as IL-6 and CRP were observed in the paediatric patients with COVID-19. The reason for this finding could be that cytokine storm plays a role in the immune pathophysiology of COVID-19.[23] Similar immune mechanisms have been reported in previous influenza studies.[24] However, it would need more data to determine if cytokine storms are a major determinant of severity and probably of fatal outcome. Furthermore, some paediatric patients with COVID-19 presented with abnormal coagulation function (elevated d-dimer and activated partial thromboplastin time levels). In contrast with the findings from a study of adult patients with COVID-19,[25] the paediatric patients with COVID-19 with these abnormalities had mostly mild or asymptomatic-type disease, and there was no obvious evidence linking abnormal coagulation function with poor prognosis. Self-controlled studies of paediatric patients with COVID-19 found significant increases in lymphocyte levels after treatment, suggesting that lymphocyte numbers are reduced in paediatric COVID-19 cases.[10] It has been suggested that this virus might directly infect lymphocytes, especially T lymphocytes, and initiate or promote lymphocyte cell death, resulting in lymphocyte depletion and a reduced antiviral response.[26]

The Cox regression model indicated that the time from positive RT-PCR to negative is strongly influenced by the presence of SARS-CoV-2-infected individuals in their family. This finding suggests that the main transmission route for COVID-19 in children is close contact with family members, which is consistent with findings from another research.[27] Therefore, social distancing and hygiene measures should be exercised in an intimate transmission scenario (eg, family settings[28]) containing a risk of higher viral load and exposure, which is also supported by the conclusions of Bielecki et al[29] and Dalton et al[30] that non-pharmaceutical measures can help reduce the probability of infection or prevent it from developing to more severe disease by quantitatively reducing the viral inoculum. We found 59 paediatric COVID-19 cases who were the first SARS-CoV-2-infected individuals in their families, which

indicated that paediatric patients with COVID-19 could be a primary source of infection, a potential causation pathway often ignored.[8 31] Because the disease severity of paediatric patients is mainly mild and asymptomatic, it might be more difficult to detect such cases in the early stage of disease. In addition, we found that the vast majority of children with symptoms of emesis were aged below 5 years old, which is a very fragile and vulnerable stage of life. On one hand, these patients need longer time to return negative RT-PCR because they are weaker to clear the virus and their immunity system is still immature. On the other hand, the literature has also shown that the occurrence of virus infection can affect gastrointestinal function and cline to gastrointestinal symptoms[32–34] (eg, emesis[19]).

This study had some limitations. First, we failed to capture all asymptomatic patients as the nature of a hospital-based study. However, this failure had limited impact on the interpretation of our study because it mainly caused an underestimation of the true prevalence of asymptomatic patients which is beyond the objective of our research. Second, to study as large of a cohort as possible, we enrolled discharged paediatric patients with COVID-19 from multiple hospitals, which can be very heterogeneous in terms of treatment and management characteristics. Possible limits due to these aspects were the use of traditional Chinese medicine and international normalise ratio of prothrombin time which made the data hardly comparable with those of other nations; and caution should be also exercised in extrapolating these results to other countries, as children in China may have different baseline health conditions and medical care access. Additionally, given this is a hospital-based study, we did not have accurate date if a patient was diagnosed before admission to hospital; the starting point of positive diagnosis was therefore defined by the date of admission. Caution should be exercised in explanation of these results.

## CONCLUSION

In conclusion, the paediatric cases with emesis symptom, complications or with infected family members will extend the length of duration from positive RT-PCR to negative. We suspected that the infecting viral load influences disease severity that emphasised on the importance of social distancing and interventions such as masks. It is difficult to detect asymptomatic paediatric patients with COVID-19; therefore, it poses a potential risk to the community.

**Author affiliations**
[1]Department of Radiological Medicine, Huazhong University of Science and Technology Tongji Medical College Affiliated Wuhan Children's Hospital, Wuhan, Hubei, People's Republic of China
[2]Department of Nephrology, Children's Hospital of Fudan University, Shanghai, People's Republic of China
[3]Department of Epidemiology, Fudan University School of Public Health, Shanghai, People's Republic of China

[4]Department of Health Economics, Fudan University School of Public Health, Shanghai, People's Republic of China

[5]Department of Pharmacy, Huazhong University of Science and Technology Tongji Medical College Affiliated Wuhan Children's Hospital, Wuhan, Hubei, People's Republic of China

[6]Department of Pediatrics, The Third People's Hospital of Shenzhen, Shenzhen, Guangdong, People's Republic of China

[7]Department of Clinical Pharmacy, Children's Hospital of Fudan University, Shanghai, People's Republic of China

[8]Department of Pharmacy, Guangzhou Women and Children's Medical Center, Guangzhou, Guangdong, People's Republic of China

[9]Department of Pharmacy, The Third People's Hospital of Kunming, Kunming, Yunnan, People's Republic of China

[10]Department of Pharmacy, Children's Hospital Affiliated to Zhengzhou University, Zhengzhou, Henan, People's Republic of China

[11]Department of Pharmacy, Guangzhou Eighth People's Hospital, Guangzhou, Guangdong, People's Republic of China

[12]Department of Pediatrics, Guizhou Provincial People's Hospital, Guiyang, Guizhou, People's Republic of China

[13]Department of Pharmacy, The Second Affiliated Hospital and Yuying Children's Hospital of Wenzhou Medical University, Wenzhou, Zhejiang, People's Republic of China

[14]Department of Pharmacy, The First Affiliated Hospital of University of Science and Technology of China, Hefei, Anhui, People's Republic of China

[15]Department of Biology, University of California Santa Barbara, Santa Barbara, California, USA

[16]Centre of Cardiovascular, Children's Hospital of Fudan University, Shanghai, People's Republic of China

**Acknowledgements** We would like to extend our sincere thanks to Professor Stuart MacLeod (University of British Columbia) for his insightful comments. We thank Katie Oakley, from Liwen Bianji, Edanz Editing China (www.liwenbianji.cn/ac), for editing the English text of a draft of this manuscript.

**Contributors** WW, GH and ZL conceptualised and designed the study, completed the literature search and critically revised the manuscript for important intellectual content. HX, XW, JL, YH, GW, YH and JC designed the data collection instruments, collected data and are responsible for the integrity of the data, reviewed and revised the manuscript. SM, SZ, YC, RC, JL and XL coordinated and supervised data collection, conducted the initial statistical analysis of epidemiology data, reviewed and revised the manuscript. JS, HX, ZL and XY constructed the Cox proportional hazards regression models and conducted the further statistical analysis of the study, and drafted the manuscript. All authors approved the final manuscript as submitted to BMJ Open with the signature of the corresponding author and agree to be accountable for all aspects of the work. WW is the guarantor for the study and accepts full responsibility for the finished work and/or the conduct of the study, had access to the data, and controlled the decision to publish.

**Funding** This study was funded by the Bill & Melinda Gates Foundation (grant number INV-006277).

**Disclaimer** The funder had no role in the study design, data collection, data analysis, data interpretation or writing of the report. The corresponding author had full access to all data in the study and had final responsibility for the decision to submit for publication.

**Map disclaimer** The inclusion of any map (including the depiction of any boundaries therein), or of any geographic or locational reference, does not imply the expression of any opinion whatsoever on the part of BMJ concerning the legal status of any country, territory, jurisdiction or area or of its authorities. Any such expression remains solely that of the relevant source and is not endorsed by BMJ. Maps are provided without any warranty of any kind, either express or implied.

**Competing interests** None declared.

**Patient and public involvement statement** Patients and/or the public were not involved in the design, or conduct, or reporting or dissemination plans of this research.

**Patient consent for publication** Not required.

**Ethics approval** This study was approved by the ethics committee of the National Children's Medical Center, Children's Hospital of Fudan University (No. (2020) 187), and followed the Declaration of Helsinki. Written informed consent was obtained from the guardians of the patients prior to conducting the study.

**Provenance and peer review** Not commissioned; externally peer reviewed.

**Data availability statement** Data are available upon reasonable request. The data that support the findings of this study are available from the corresponding author, Weibing Wang, upon reasonable request.

**ORCID iDs**
Zhixi Liu http://orcid.org/0000-0002-8552-4027
Weibing Wang http://orcid.org/0000-0002-4497-5251

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
