## [Reviewer comments · BMJ Open]

ARTICLE DETAILS

TITLE (PROVISIONAL)	Factors associated with the time to return negative RT-PCR from COVID-19 in pediatric patients: a retrospective cohort study
AUTHORS	Shao, Jianbo; Xu, Hong; Liu, Zhixi; YING, Xiaohua; Xu, Hua; Wang, Xianfeng; Lu, Jinmiao; Huang, Yidie; Wang, Guangfei; He, Yanling; Chen, Jie; Ma, Shuli; Zou, Shangrong; Cui, Yuxia; Chen, Ruijie; Lu, Jin; Li, Xuyuan; Li, Zhiping; Huang, Guoying; Wang, Weibing

VERSION 1 – REVIEW

REVIEWER	Reo Tanoshima Yokohama City University, Department of Pediatrics
REVIEW RETURNED	25-May-2021

GENERAL COMMENTS	This study describes the results of retrospective observational study on clinical characteristics and factors associated with the time to negative COVID-19 in pediatric patients. The main finding of this study is that the presence of confirmed patients in the family and patients with gastrointestinal symptoms are associated with longer time to achieve negative PT-PCR. This is an interesting and important epidemiological study, but I would like the authors to clarify several things before further decision. 1. Please explain the age range of the patients including in the study. This is critical in pediatric study.2. The definition of abnormal findings such as myocardial injury or liver sufficiency are not clear. Please describe the definitions in the methods.3. Because the patients of this study are tested by either suspected symptoms or potential exposure, the authors did not capture all (especially asymptomatic) COVID-19 patients in the community. Please describe how this impacts the interpretation of this study.4. Despite the previous reports by Bielecki et al, I still do not understand why intimate transmission is associated with severe disease. The severity of the disease is determined by virulence and host factors. Minor comments Overall: Please correct grammatical errors. Page 5 Line 6: Please add COVID-10 after Coronavirus disease 2019 using parenthesis. Page 6 Line 28: Please explain what data the authors collect instead of stating 'routine tests'. Table 2. Please use international normalize ration of PT if available.
---

REVIEWER	Cinzia Auriti Ospedale Pediatrico Bambino Gesù, Neonatology
REVIEW RETURNED	06-Jun-2021

GENERAL COMMENTS	The authors analyze the possible factors associated with the negative timing of viral shedding in COVID 19 patients in China. Just some remarks Page 4 line 14 to 15 Abstract Authors: this retrospective study examined 233 pediatric cases confirmed COVID-19 Reviewer: Please change in “This retrospective study examined 233 clinical charts of pediatric patients with confirmed diagnosis of COVID 19 admitted Lines 17 to 20 Authors:medical records of these cases,.... Reviewer: please change in “medical records of these patients”... Lines 27 to 28 Authors: The median age of the cohort was.... Reviewer: please change in “The median age of patients in our cohort was.....” Lines 33 to 35 Authors: “In Cox regression analysis, longer time to negative RT-PCR was associated with the presence of confirmed patients in the family...” Reviewer: Please change in “In Cox regression analysis, longer time to negative RT-PCR was associated with the presence of confirmed infection in family members “ Authors: “Patients with gastrointestinal symptoms, complications or infected members in families may prolong rehabilitation.” Reviewer: Please change in “Patients with gastrointestinal symptoms, complications or confirmed infection in family members may have longer healing times than others. “” Authors: “However, no significant favorable effects from TCM when the patients have received antiviral treatment”. Reviewer: Please what TCM does mean?? Please expand the acronym. Limitations Authors: “This study was limited to a small number of pediatric COVID-19 patients, given that the prevalence of COVID-19 pediatric cases is relatively low compared with adults population”. Reviewer: I don't think it's a limitation but a strength of the study. If the prevalence in children is low (thankfully) a retrospective study is ideal. The limit may be that the cases scattered over 12 hospitals in different cities can be very heterogeneous in terms of diagnosis and treatment characteristics, as there are no validated protocols on the subject, given the novelty. We all do what we can do. Please delete this limitation, also from the end of the paper and discuss on heterogeneity of management. Possible limits due to this aspect is the use of Chinese medicine, which makes the data hardly comparable with those of other nations. The worst part of the manuscript is the abstract, to be corrected as suggested for the syntax. The rest of the paper is much better written and interesting. The figures and tables are understandable and in an acceptable number. Therefore by modifying the abstract as suggested the work can be published. The problem of submission to the ethics committee is unclear. For retrospective studies, the Ethics Committee should be notified of the ongoing study.
---

	I am available for the further revision
--	---

REVIEWER	Tumena Corrah London North West University Healthcare NHS Trust, Infectious Diseases
REVIEW RETURNED	14-Jun-2021

GENERAL COMMENTS	I think the standard of the written English could be better. The authors list the factors associated with time to negative covid pcr as having family members with a positive covid test, presence of vomiting/emesis and having severe disease. We know that some patients with covid remain PCR positive for lengthy duration - these patients are not infective and positive result likely reflects dead virus. I cannot find biological plausibility why vomiting should be associated with time to negative PCR result. Furthermore, in their definition of severity of disease (supplementary table 1), emesis is present in patients with both mild and severe disease. I do not think this paper adds anything new to literature.
--

REVIEWER	Justus Simba Jomo Kenyatta University of Agriculture and Technology, Child Health and Paediatrics
REVIEW RETURNED	23-Jun-2021

GENERAL COMMENTS	I was privileged to review your work. I have the following comments for consideration:  1. Throughout the document, it is repeated that those with gastrointestinal symptoms had "prolonged time to return to negative". However, looking at the analysis it appears this only applies to emesis and not diarrhoea. This needs clarification. 2. Abstract Page 5 line: 38-43 starting with "Usage.." is unclear, kindly revise to make meaning clear. 3. Introduction Page 6, line: 7-9, 'As of 29th June 2020' why a date a year ago? This should be updated to a more recent point. 4. Methods, Page 7 line 12, the authors assert, "written consent was taken". This being a retrospective data, at what point was this taken? 5. In page 13, line 14 lymphocytes percentage is 229; while the whole blood cells is 227 patients, did you have patients with lymphocyte % but no WBC count? 6. Most of the references are up to May 2020 (one year ago), this can be updated as a lot has changed as pertains COVID-19. Thank you
--

VERSION 1 – AUTHOR RESPONSE

Reviewer: 1

Dr. Reo Tanoshima, Yokohama City University

Comments to the Author:

Comment 1: This study describes the results of retrospective observational study on clinical characteristics and factors associated with the time to negative COVID-19 in pediatric patients. The main finding of this study is that the presence of confirmed patients in the family and patients with gastrointestinal symptoms are associated with longer time to achieve negative PT-PCR. This is an interesting and important epidemiological study, but I would like the authors to clarify several things before further decision.

Response: We appreciate your generous comments on our manuscript.

Comment 2: Please explain the age range of the patients including in the study. This is critical in pediatric study.

Response: Thank you. As suggested, we have added the age range of the patients included in our study in the corresponding part. Please see page 9 line 52-54.

Comment 3: The definition of abnormal findings such as myocardial injury or liver insufficiency are not clear. Please describe the definitions in the methods.

Response: Thank you for pointing out it. As suggested, we have added the definitions of myocardial injury and liver insufficiency in the corresponding part of methods. Please see page 7 line 26-31.

Comment 4: Because the patients of this study are tested by either suspected symptoms or potential exposure, the authors did not capture all (especially asymptomatic) COVID-19 patients in the community. Please describe how this impacts the interpretation of this study.

Response: Thank you for your opinion and we agree with your point. We failed to capture all asymptomatic patients for these two reasons:

a) the study was conducted based on hospitals which relied on passive reporting of patients by themselves;

b) the patients without symptoms were much less likely to seek healthcare service than those with symptoms unless there was mass screening or became a close contact.

Although this likely causes an underestimation of the true prevalence of asymptomatic SARS-CoV-2 infection, estimation the prevalence isn't the purpose of our research. Besides, the feature of asymptomatic patients is without any clinical symptoms and no evidence exhibits asymptomatic patients need a longer time to return negative RT-PCR than other clinical types at present. Therefore, the failure to capture all asymptomatic patients have limited impact on the interpretation of our study. We have added this as a limitation in the "discussion" section. Please see page 20 line 4-15.

Comment 5: Despite the previous reports by Bielecki et al, I still do not understand why intimate transmission is associated with severe disease. The severity of the disease is determined by virulence and host factors.

Response: Thank you for the question. We agree that severity of the disease is determined by virulence and host factors. We are sorry for the misunderstanding and have corrected the inappropriate descriptions given that our study is unable to directly conclude that intimate transmission is associated with severe disease.

The Bielecki *et al.*¹ provided evidence that social distancing and hygiene measures had been shown to quantitatively reduce the viral inoculum during infection. Another study also suggested lower intensity of viral exposure might be another factor leading to less severe disease². Besides, given that the SARS-CoV-2 is transmitted through contact, droplets, or aerosols, if there is an infected patient in the family environment and without any quarantine measures, the objects or spaces frequently touched and stayed by this infected patient will remain a high viral load and exposure which could bring greater health risks and threats to family members in closely contact with the space.

Therefore, strict non-pharmaceutical measures should be implemented in intimate transmission scenario with confirmed COVID-19 patient, especially in family settings, to reduce the probability of

infection or prevent developing to more severe disease. The details of revision are shown in page 19 line 9-25.

Minor comments

Comment 7: Overall: Please correct grammatical errors.

Response: We appreciate your generous comments on our manuscript. Please see page 16 Table 3, page 19 line 9.

Comment 8: Page 5 Line 6: Please add COVID-19 after Coronavirus disease 2019 using parenthesis.

Response: Thank you for pointing out this. As suggested, we have revised them. Please see page 6 line 6 and 15.

Comment 9: Page 6 Line 28: Please explain what data the authors collect instead of stating 'routine tests'.

Response: Thank you for pointing out this. After check, we confirmed the data of routine tests collected is blood routine examination data. Therefore, we have replaced the "routine tests" with "blood routine examination ". Please see page 7 line 33-34.

Comment 10: Table 2. Please use international normalize ration of PT if available.

Response: Thanks for your good suggestion. We agree that using international normalize ration of PT would be better to correct the difference of thrombin reagents and make the measured results comparable. Besides, the calculation method of the international normalized ratio (INR) is the ISI power of the ratio of prothrombin time of patients and normal controls. However, we didn't collect the international sensitivity index (ISI) of each reagent which is determined by the manufacturer upon delivery. Therefore, we are so sorry that we can't provide the international normalize ration of PT. Besides, we have included this point as a limitation. Please see page 20 line 19-25.

Reviewer: 2

Dr. Cinzia Auriti, Ospedale Pediatrico Bambino Gesù

Comments to the Author:

Comment 1:

Authors: this retrospective study examined 233 pediatric cases confirmed COVID-19

Reviewer: Please change in "This retrospective study examined 233 clinical charts of pediatric patients with confirmed diagnosis of COVID 19 admitted"

Response: We appreciated this suggestion. We have revised this sentence as suggested. Please see page 3 line 17-20.

Comment 2: Lines 17 to 20

Authors:medical records of these cases,....

Reviewer: please change in "medical records of these patients"...

Response: Thank you. We have revised this sentence as suggested. Please see page 3 line 35-36.

Comment 3: Lines 27 to 28

Authors: The median age of the cohort was....

Reviewer: please change in "The median age of patients in our cohort was....."

Response: Thank you. We have revised this sentence as suggested. Please see page 3 line 45-46.

Comment 4: Lines 33 to 35

Authors: "In Cox regression analysis, longer time to negative RT-PCR was associated with the presence of confirmed patients in the family..."

Reviewer: Please change in "In Cox regression analysis, longer time to negative RT-PCR was associated with the presence of confirmed infection in family members "

Response: We appreciated this suggestion. We have revised this sentence as suggested. Please see page 3 line 50-54.

Comment 5:

Authors: "Patients with gastrointestinal symptoms, complications or infected members in families may prolong rehabilitation."

Reviewer: Please change in "Patients with gastrointestinal symptoms, complications or confirmed infection in family members may have longer healing times than others. ""

Response: We appreciated this suggestion. We have revised this sentence as suggested. Please see page 4 line 11-15.

Comment 6:

Authors: "However, no significant favorable effects from TCM when the patients have received antiviral treatment".

Reviewer: Please what TCM does mean?? Please expand the acronym.

Response: Thanks for pointing out it. TCM is the acronym of traditional Chinese medicine. Besides, we have added this in the corresponding part. Please see page 3 line 58-59.

Comment 7: Limitations

Authors: "This study was limited to a small number of pediatric COVID-19 patients, given that the prevalence of COVID-19 pediatric cases is relatively low compared with adults population".

Reviewer: I don't think it's a limitation but a strength of the study. If the prevalence in children is low (thankfully) a retrospective study is ideal. The limit may be that the cases scattered over 12 hospitals in different cities can be very heterogeneous in terms of diagnosis and treatment characteristics, as there are no validated protocols on the subject, given the novelty. We all do what we can do.

Please delete this limitation, also from the end of the paper and discuss on heterogeneity of

management. Possible limits due to this aspect is the use of Chinese medicine, which makes the data hardly comparable with those of other nations.

Response: Thanks for your advice. As suggested, we have deleted this limitation and added the discussion on the heterogeneity of management as a limitation. The details showed in page 5 line 17-20 and page 20 line 14-30.

Comment 8: The worst part of the manuscript is the abstract, to be corrected as suggested for the syntax. The rest of the paper is much better written and interesting. The figures and tables are understandable and in an acceptable number. Therefore, by modifying the abstract as suggested the work can be published.

Response: Thanks for your comments and advice. We have tried our best to revise the abstract. Please see page 3-4.

Comment 9: The problem of submission to the ethics committee is unclear. For retrospective studies, the Ethics Committee should be notified of the ongoing study.

Response: Thank you for pointing out this. As suggested, we have added the "Ethics approval" section in the required place. Please see page 7 line 6-12, page 21 line 57-60 and page 22 line 4-7.

Comment 10: I am available for the further revision.

Response: Thank you for the comment.

Reviewer: 3

Dr. Tumena Corrah, London North West University Healthcare NHS Trust

Comments to the Author:

Comment 1: I think the standard of the written English could be better.

Response: Thank you for the suggestion. As suggested, we have revised the written English of our manuscript shown in red words.

Comment 2: The authors list the factors associated with time to negative covid pcr as having family members with a positive covid test, presence of vomiting/emesis and having severe disease. We know that some patients with covid remain PCR positive for lengthy duration - these patients are not infective and positive result likely reflects dead virus. I cannot find biological plausibility why vomiting should be associated with time to negative PCR result. Furthermore, in their definition of severity of disease (supplementary table 1), emesis is present in patients with both mild and severe disease.

Response: Thank you for the comments. In fact, we found that 86% of the children with emesis symptom are under the age of 5 years old, which is a very fragile and vulnerable stage of life. On the one hand, these patients need longer time to turn negative RT-PCR may be due to that they are weaker to clear the virus and their immunity system is still immature. On the other hand, the literature

has also shown that the occurrence of virus infection can affect gastrointestinal function and prone to gastrointestinal symptoms³⁻⁵. This could explain what we found at a certain degree.

In addition, given that there is no direct causal relationship between the time to turn negative and the severity of the disease, it's reasonable that emesis symptom is present in patients with both mild and severe disease, which is consistent with our data.

We have revised the descriptions of corresponding part. The details are shown in page 19 line 32-59.

Reviewer: 4

Dr. Justus Simba, Jomo Kenyatta University of Agriculture and Technology

Comments to the Author:

Comment 1: Throughout the document, it is repeated that those with gastrointestinal symptoms had "prolonged time to return to negative". However, looking at the analysis it appears this only applies to emesis and not diarrhea. This needs clarification.

Response: Thank you for the suggestion. As suggested, we have corrected the inappropriate descriptions by replacing the gastrointestinal symptoms with emesis symptom. Please see page 3 line 56, page 4 line 11-12, page 17 line 30 and page 20 line 36-37.

Comment 2: Abstract Page 5 line: 38-43 starting with "Usage.." is unclear, kindly revise to make meaning clear.

Response: Thank you for pointing out this. As suggested, we have revised the sentence to be clearer and more understandable. Please see page 3 line 58-59 and page 4 line 4.

Comment 3: Introduction Page 6, line: 7-9, 'As of 29th June 2020' why a date a year ago? This should be updated to a more recent point.

Response: Thank you for pointing out this. As suggested, we have updated the data to the latest point. Please see page 6 line 6-12.

Comment 4: Methods, Page 7 line 12, the authors assert, "written consent was taken". This being a retrospective data, at what point was this taken?

Response: Thank you for asking. We have supplemented the time point of taking written consent of patients. The details show in page 22 line 4-7.

Comment 5: In page 13, line 14 lymphocytes percentage is 229; while the white blood cells are 227 patients, did you have patients with lymphocyte % but no WBC count?

Response: Thank you for pointing out this mistake. After verifying the original database, it was found that the two data of white blood cell counts were omitted in the statistical process of Table 2. Now the corresponding results and descriptions of this part have been recalculated and corrected. Although the mistake had little influence on the original results, we are deeply sorry for this mistake. The details are shown in page 12 line 6-7 and page 13 Table 2.

Comment 6: Most of the references are up to May 2020 (one year ago), this can be updated as a lot has changed as pertains COVID-19. Thank you.

Response: Thank you for pointing out this. As suggested, we have updated the references. Please see page 6 line 11-12, page 17 line 14-15 and 22-23, page 19 line 9-10, 30 and 45-46.

VERSION 2 – REVIEW

REVIEWER	Reo Tanoshima Yokohama City University, Department of Pediatrics
REVIEW RETURNED	29-Aug-2021

GENERAL COMMENTS	This is a revised manuscript describing the results of retrospective observational study on clinical characteristics and factors associated with the time to negative COVID-19 in pediatric patients. I think the authors mostly described my previous questions and comments. My final comment is about liver insufficiency, which the authors added in line 9 of page 7. 'the inability of the liver to perform its normal synthetic and metabolic functions, etc.' is still vague. Please state specific criteria to define liver dysfunction.
---

REVIEWER	Tumena Corrah London North West University Healthcare NHS Trust, Infectious Diseases
REVIEW RETURNED	26-Aug-2021

GENERAL COMMENTS	Improvement from initial manuscript. I still find article somewhat difficult to read/follow due to grammatical errors which are fixable. I would appreciate further clarification on one point and believe authors should address this prior to publication. Authors state length of time from positive RT-PCR to negative RT-PCR ranged from 0 to 32 days. For those individuals with 0 days between positive and negative swabs , i.e they had positive and negative swabs taken on the same day, could the initial positive RT-PCR be a false positive? How do the authors explain this? Does eliminating those individuals alter the results?
---

REVIEWER	Justus Simba Jomo Kenyatta University of Agriculture and Technology, Child Health and Paediatrics
REVIEW RETURNED	07-Sep-2021

GENERAL COMMENTS	The authors have addressed issues as raised by reviewers adequately.
--

VERSION 2 – AUTHOR RESPONSE

Reviewer: 1

Dr. Reo Tanoshima, Yokohama City University

Comments to the Author:

Comment 1: This is a revised manuscript describing the results of retrospective observational study on clinical characteristics and factors associated with the time to negative COVID-19 in pediatric

patients. I think the authors mostly described my previous questions and comments.

My final comment is about liver insufficiency, which the authors added in line 9 of page 7. 'The inability of the liver to perform its normal synthetic and metabolic functions, etc.' is still vague. Please state specific criteria to define liver dysfunction.

Response: Thank you for pointing out it. As suggested, we have added the specific criteria of liver dysfunction definition in page 7 line 26-31.

Reviewer: 3

Dr. Tumena Corrah, London North West University Healthcare NHS Trust

Comments to the Author:

Comment 1: I still find article somewhat difficult to read/follow due to grammatical errors which are fixable.

Response: Thank you for pointing out it. As suggested, we have re-checked the article and tried our best to correct the grammatical errors. Please see page 6 line 19-20, page 7 line 36-37 and page 19 line 17-25.

Comment 2: I would appreciate further clarification on one point and believe authors should address this prior to publication. Authors state length of time from positive RT-PCR to negative RT-PCR ranged from 0 to 32 days. For those individuals with 0 days between positive and negative swabs, i.e they had positive and negative swabs taken on the same day, could the initial positive RT-PCR be a false positive? How do the authors explain this? Does eliminating those individuals alter the results?

Response: Thank you for arising this important issue. In fact, the existence of individuals with 0 day between positive and negative result can probably be explained that: This is a hospital-based study. As we did not have accurate date if a patient was diagnosed before admission to hospital, the starting point of positive diagnosis was therefore defined by the data of admission. We agree that this is a limitation and have clarified in the limitation section of the manuscript. The details showed in page 20 line 4-10.

However, the discharge criteria of each pediatric patient should meet two consecutive negative RT-PCR test results. Usually, the patient who have a positive RT-PCR before admission would have a second test to confirm the diagnosis, which reduce the possibility of false positivity. In addition, related literatures¹ have evaluated the nucleic acid detection methods which are commonly used in clinical practices; and found that the specificity results of these methods are as high as 100%. Therefore, the possibility of false positives is extremely low.

Finally, we did additional analysis based on removing the data with the negative time of 0 (three observations) and the results were showed below (**Table R1**). There were no substantially changes in the scenario to remove data of 0, which proved the reliability of the original results.

Table R1. Hazard Ratios of Time to Return Negative RT-PCR from COVID-19 in Cox Proportional Hazards Regression Model. (Based on removing the data with the negative time of 0)

Risk factors	Univariate Cox regression ^a		Multivariate Cox regression ^{a, b}	
	HR (95% CI)	p	HR (95% CI)	p
Clinical characteristics				
Sex (versus male)	1.00 (0.80-1.35)	0.774	-	-
Age	0.84 (0.64-1.10)	0.199	-	-
Contact history	1.50 (1.01-2.17)	0.043	1.12(0.72-1.73)	0.629
Clinical symptoms				
Fever	0.96 (0.73-1.26)	0.773	0.99(0.74-1.32)	0.947
Cold	1.10 (0.28-4.51)	0.874	1.16(0.29-4.71)	0.835
Diarrhea	1.80 (0.99-3.21)	0.053	1.61(0.87-2.99)	0.128
Emesis	0.51 (0.24-1.12)	0.093	0.34(0.14-0.80)	0.014
Nausea	0.80 (0.56-1.15)	0.230	0.72(0.49-1.04)	0.083
Running nose	1.00 (0.63-1.56)	0.983	0.97(0.59-1.56)	0.875
Coughing of phlegm	0.95 (0.62-1.47)	0.827	0.91(0.58-1.45)	0.702
Complications	0.74 (0.51-1.10)	0.136	0.58(0.37-0.90)	0.015
Clinical type (versus asymptomatic)				
Mild	0.92(0.65-1.30)	0.642	0.94(0.66-1.34)	0.735
Moderate	0.94(0.57-1.55)	0.815	0.89(0.52-1.53)	0.685
Severe	0.30(0.10-0.87)	0.030	0.14(0.03-0.61)	0.009
Household contact				
Family member infection	0.65 (0.48-0.89)	0.007	0.56(0.40-0.79)	0.001
Number of infectious family member				
1	0.57(0.40-0.81)	0.002	0.47(0.32-0.69)	<0.001
2	0.63(0.43-0.91)	0.014	0.52(0.35-0.77)	0.001

≥3	0.59(0.39-0.89)	0.012	0.54(0.35-0.83)	0.005
Other respiratory pathogens infection				
Combined with virus ^c infection	0.87 (0.61-1.25)	0.457	0.80(0.56-1.16)	0.235
Combined with bacterial infection ^d	1.10 (0.80-1.42)	0.659	1.02(0.76-1.38)	0.875
Treatment (versus ATD)				
TCM ^e +ATD ^f	0.81 (0.62-1.06)	0.124	0.87(0.65-1.15)	0.317

Reviewer: 4

Dr. Justus Simba, Jomo Kenyatta University of Agriculture and Technology
Comments to the Author:

Comment 1:

The authors have addressed issues as raised by reviewers adequately.

Response: We appreciate your comments on our manuscript.

VERSION 3 – REVIEW

REVIEWER	Reo Tanoshima Yokohama City University, Department of Pediatrics
REVIEW RETURNED	18-Sep-2021
GENERAL COMMENTS	The authors appropriately revised the manuscripts with response to the reviewers' comments.
REVIEWER	Tumena Corrah London North West University Healthcare NHS Trust, Infectious Diseases
REVIEW RETURNED	06-Oct-2021
GENERAL COMMENTS	Issues raised by reviewers addressed. However several typos remain